# Inflammatory Biomarkers for Thrombotic Risk Assessment in Multiple Myeloma Patients on IMiD/aCD38-Based Regimens: Insights from a Prospective Observational Study

**DOI:** 10.3390/biom15111533

**Published:** 2025-10-31

**Authors:** Cirino Botta, Anna Maria Corsale, Claudia Cammarata, Fabiana Di Fazio, Emilia Gigliotta, Andrea Rizzuto, Manuela Ingrascì, Maria Speciale, Cristina Aquilina, Marta Biondo, Andrea Romano, Mariasanta Napolitano, Marta Mattana, Sergio Siragusa

**Affiliations:** 1Department of Health Promotion, Mother and Child Care, Internal Medicine and Medical Specialties (ProMISE), University of Palermo, 90127 Palermo, Italy; claudiacammarata9@gmail.com (C.C.); difaziofabiana@gmail.com (F.D.F.); emiliagigliotta@gmail.com (E.G.); andrerizzuto@gmail.com (A.R.); manuelagiuseppa.ingrasci@you.unipa.it (M.I.); maria.speciale@unipa.it (M.S.); cristina.aquilina@unipa.it (C.A.); andrea.romano@unipa.it (A.R.); mariasanta.napolitano@unipa.it (M.N.); marta.mattana@unipa.it (M.M.); sergio.siragusa@unipa.it (S.S.); 2Department of General Surgery and Medical-Surgical Specialties, Hematology Section, University of Catania, Catania 95123, Italy; marta.biondo@unict.it

**Keywords:** multiple myeloma, thrombosis, immunomodulatory drugs (IMiDs)

## Abstract

Thrombosis is a common complication in multiple myeloma (MM) patients treated with immunomodulatory drugs (IMiDs), including thalidomide, lenalidomide, and pomalidomide. When combined with anti-CD38 monoclonal antibodies, these agents are highly effective but may increase thrombotic events (TE), potentially delaying therapy. This exploratory, hypothesis-generating analysis, conducted within the MMVision mono-institutional prospective study, included 53 MM patients who initiated IMiD plus anti-CD38 therapy between May 2021 and December 2022 (median follow-up: 18 months). Treatment regimens comprised lenalidomide (n = 36) or thalidomide (n = 15) with daratumumab, and pomalidomide (n = 2) with isatuximab. Most patients (n = 38) received frontline therapy, and all were given thromboprophylaxis according to guidelines, mainly aspirin (73%). Five patients (9.4%) developed VTE after a median of 48 days, managed with short-term low-molecular-weight heparin (LMWH). Exploratory analysis of 27 clinical/laboratory parameters suggested possible associations between VTE and low levels of beta-2 microglobulin, ferritin, intact/free lambda light chains, and monocyte-to-lymphocyte ratio. Notably, four of the five VTEs occurred in patients without lytic bone disease, typically associated with bone-driven inflammation in MM. Although all patients received aspirin prophylaxis from treatment initiation, it remains unclear whether thrombosis would also have occurred among those with higher inflammatory burden. These preliminary observations may indicate that in patients with relatively lower inflammation, aspirin prophylaxis could be less effective, potentially favoring VTE onset. In two VTE cases, cytokine profiling showed decreased M-CSF, SCLF-β, and MIP-1α, with increased G-CSF, raising the hypothesis of distinct immune-inflammatory pathways contributing to TEs. Given the limited number of patients and thrombotic events, and the cytokine data available for only two VTE cases, these associations should be regarded as exploratory and interpreted with caution. Overall, these exploratory findings warrant validation in larger, independent cohorts and may help generate hypotheses on how inflammatory signatures influence thrombotic risk and prophylaxis efficacy in MM patients receiving IMiD/anti-CD38-based regimens.

## 1. Introduction

Multiple myeloma (MM) is a malignant plasma cell disorder characterized by bone marrow infiltration, end-organ damage, and immune dysregulation. Despite major therapeutic advances with proteasome inhibitors, immunomodulatory drugs (IMiDs), and anti-CD38 monoclonal antibodies, MM still remains incurable, with patients requiring continuous treatment [1]. These regimens, while highly effective, are associated with increased thrombotic risk.

Thrombotic events (TE), indeed, represent a significant and potentially life-threatening complication in patients with MM, particularly those undergoing treatment with IMiDs such as thalidomide (Thal), lenalidomide (Len), and pomalidomide (Pom) [2,3,4,5], essential agents for MM therapy with a remarkable efficacy when combined with anti-CD38 monoclonal antibodies [6,7,8]. Venous thromboembolism (VTE) occurs with a reported incidence rate exceeding 10% over MM course [9]. Notably, the combination of IMiDs with high-dose corticosteroids and other chemotherapeutic agents can further increase the risk, with thrombosis rates reaching approximately 26% in specific treatment settings [10,11,12,13]. Although anti-CD38 antibodies themselves have not been associated with an increased risk of venous thromboembolism, their combination with IMiDs and high-dose corticosteroids may amplify patient-related thrombotic susceptibility [14]. Thrombotic complications not only contribute to morbidity and mortality of MM patients but they also interfere with treatment continuity, often necessitating treatment schedule variations or discontinuation [15]. Consequently, understanding the underlying mechanisms of VTE in MM, identifying reliable predictive biomarkers, and optimizing thromboprophylaxis strategies are crucial for improving MM global care and patient outcomes.

The pathophysiology of thrombosis in MM is multifactorial and involves both disease-related and treatment-related factors [5]. MM itself is a highly prothrombotic malignancy due to interactions between malignant plasma cells, the bone marrow microenvironment, and systemic inflammatory mediators [16,17,18]. These interactions promote inflammation, endothelial dysfunction, platelet activation, and coagulation abnormalities—core elements of the classical Virchow’s triad. Elevated pro-inflammatory cytokines such as interleukin-6 (IL-6), tumor necrosis factor-α (TNF-α), and interleukin-1β (IL-1β) contribute to endothelial activation by enhancing adhesion molecule expression, tissue factor release, and platelet reactivity [19]. Collectively, these processes establish a persistent hypercoagulable state that may be further amplified by therapeutic agents. In addition, increased circulating microparticles and altered fibrin network architecture have been reported in MM, linking immune activation to thrombin generation and impaired fibrinolysis [9,20].

IMiDs further exacerbate thrombotic risk through multiple mechanisms, including increased cytokine release, upregulation of endothelial adhesion molecules, and promotion of a hypercoagulable state. Thalidomide, lenalidomide, and pomalidomide enhance T-cell and NK-cell activity while downregulating regulatory T cells, leading to secondary cytokine surges involving IL-2, IL-6, and interferon-γ. In addition, IMiDs can directly influence endothelial homeostasis by inducing phosphatidylserine exposure, reducing nitric oxide bioavailability, and increasing expression of vascular cell adhesion molecule-1 (VCAM-1). These biological perturbations may amplify the baseline prothrombotic tendency inherent to MM, potentially explaining the high incidence of venous thromboembolism observed in patients exposed to IMiD-containing regimens.

Emerging evidences also suggest that monoclonal antibodies such as daratumumab and isatuximab may exert variable effects on thrombotic risk, warranting further investigation into their role in coagulation dynamics [21,22].

Given the clinical burden of thrombosis in MM, the development of effective risk stratification models and evidence-based thromboprophylaxis guidelines remains an urgent priority. In this context, an expert panel recently conducted a comprehensive review of the literature, proposing a structured framework to optimize thrombosis prevention in MM patients receiving active treatment [2]. This framework encompasses four key domains: identification of thrombotic risk factors, individualized risk stratification, selection of appropriate thromboprophylaxis strategies, and management of acute and recurrent thrombotic events. Implementing these recommendations in clinical practice could significantly reduce thrombotic complications and improve adherence to MM treatment protocols.

In this context, current risk assessment tools often fall short in real-world MM populations, where data heterogeneity and evolving treatment regimens complicate patient stratification. There is growing interest in incorporating biological markers—including inflammatory and immune mediators—into predictive models to enhance their precision and applicability.

This study, conducted within the framework of the MMVision mono-institutional observational program, aims to explore inflammatory and immune-related mechanisms associated with thrombotic events in MM patients treated with IMiD- and anti-CD38-based regimens. By integrating clinical, hematological, and immune profiling data, this proof-of-concept investigation seeks to identify potential biomarkers of thrombotic risk and provide preliminary evidence supporting the development of personalized thromboprophylaxis strategies.

## 2. Materials and Methods

### 2.1. Patient Population

This study is part of a single-center prospective and retrospective study approved by our internal ethical committee with the numbers 02/2022, codename: MMVision, and from its biological counterpart approved by ethical committee with the number 05/2021. Specifically, the MMvision study aims to determine whether the optimization and integration of current clinical characterization strategies can be leveraged to monitor and predict disease evolution (from MGUS to SMM to MM) as well as treatment resistance or refractoriness, relapse, therapy suitability, outcomes, and treatment-related toxicities. A unified multiparametric database will integrate clinical, biological, and outcome data. Within this part, dedicated to adverse events, we involved 53 Newly Diagnosed Multiple Myeloma (NDMM) patients treated at our institution, “Paolo Giaccone” University Hospital of Palermo, who initiated treatment with IMiD and anti-CD38 monoclonal antibodies between May 2021 and December 2022. Additionally, all treatment regimens included weekly dexamethasone (median dose 20 mg, range 20–40 mg), adjusted for age and comorbidities in selected patients.

The median follow-up period was 18 months. Patients were excluded from the analysis if they: (1) did not sign the informed consent; or (2) had any other pathology or medical condition that, in the opinion of the investigator, could interfere with adherence to the protocol or with the expression of informed consent. Thromboprophylaxis was administered according to IMWG and institutional recommendations, based on clinical risk assessment rather than formal score application. All patients who received a IMiDs + aCD38 therapy received low-dose aspirin at treatment initiation, irrespective of their baseline risk profile. Patients who were already under anticoagulant therapy (e.g., low-molecular-weight heparin or direct oral anticoagulants) for pre-existing indications continued their ongoing regimen without modification.

### 2.2. Data Collection

Baseline demographic and clinical data were collected from medical records. The indicators retrieved included: (1) demographic characteristics such as age and baseline diseases, as well as medication history including treatment with anticoagulants or antiplatelet agents; (2) type of MM and staging; (3) laboratory data including white blood cells, hemoglobin, hematocrit, platelets, serum creatinine, albumin, globulin, calcium, lactate dehydrogenase, ferritin, beta-2 microglobulin (β2M), neutrophil/lymphocyte ratio, monocyte/lymphocyte ratio (MLR) and “First level” coagulation assays (INR, aPTT and fibrinogen; however, baseline values were unavailable for 3 out of 5 patients in the thrombosis group; for this reason, these data are presented in Appendix A as descriptive boxplots). Additionally, the study recorded thrombotic event incidence, the time to occurrence, and the clinical response to treatment. VTE were instrumentally confirmed according to current guidelines

### 2.3. VTE Risk Assessment

Four validated VTE risk assessment models were considered: the International Myeloma Working Group (IMWG) model [23], the IMPEDE VTE score [24], the SAVED score [25], and the PRISM score [26]. Each score was applied according to the original published criteria. Patients were stratified into risk categories (low, intermediate, or high) for each applicable score. Patients with missing parameters were considered unclassifiable for the corresponding score. The incidence of VTE was then analyzed in relation to these risk groupings. A descriptive comparative approach was used to evaluate the distribution of VTE cases across risk categories and assess the real-world applicability and discriminative capacity of each model.

### 2.4. Multiplex Analysis of Plasma Cytokines and Chemokines

Plasma levels of circulating and bone marrow cytokines, chemokines, and growth factors were measured using the Bio-Plex Pro™ Human Cytokine Screening Panel, 48-Plex (cat. #12007283, Bio-Rad Laboratories, Hercules, CA, USA). This comprehensive panel includes a wide range of pro- and anti-inflammatory cytokines, chemokines, as well as key growth factors and immunoregulatory molecules [FGF basic, Eotaxin, G-CSF, GM-CSF, IFN-γ, IL-1β, IL-1ra, IL-1α, IL-2Rα, IL-3, IL-12 (p40), IL-16, IL-2, IL-4, IL-5, IL-6, IL-7, IL-8, IL-9, GRO-α, HGF, IFN-α2, LIF, MCP-3, IL-10, IL-12 (p70), IL-13, IL-15, IL-17A, IP-10, MCP-1 (MCAF), MIG, β-NGF, SCF, SCGF-β, SDF-1α, MIP-1α, MIP-1β, PDGF-BB, RANTES, TNF-α, VEGF, CTACK, MIF, TRAIL, IL-18, M-CSF, TNF-β]. The multiplex assay was performed according to the manufacturer’s instructions. In summary, 50 µL of magnetic capture bead mixture, containing a unique bead set for each of the 48 analytes, was added to each well of a flat-bottom 96-well filter plate. Beads were washed twice with 100 µL of Bio-Plex wash buffer using a handheld magnetic plate washer to remove storage buffer. Next, 50 µL of either plasma sample, controls, or standard was added to the appropriate wells. Plate was sealed and incubated for 30 min at room temperature on a plate shaker at 850 rpm, protected from light to prevent fluorophore degradation. Following incubation, beads were washed three times to remove unbound proteins. Subsequently, 25 µL of 1X biotinylated detection antibody was added to each well, and the plate was incubated for 30 min at room temperature under the same shaking conditions. Another washing step (three washes) was then performed. Afterward, 50 µL of 1X streptavidin-phycoerythrin (SA-PE) solution was added to each well and incubated for 10 min at room temperature in the dark, with shaking. A final wash step (three washes) was conducted, and beads were resuspended in 125 µL of assay buffer and shaken for 30 s to ensure homogeneity before reading. Data acquisition was carried out using the Bio-Plex 200 System (Bio-Rad Laboratories Inc., Hercules, CA, USA) equipped with Luminex xMAP technology, and data were collected and analyzed by the Bio-Plex Manager Software version 6.0 (Bio-Rad Laboratories Inc., Hercules, CA, USA).

### 2.5. Statistical Analysis

Continuous data are shown as mean ± standard deviation or median. Categorical variables are presented as numbers or percentages. Differences between the groups were assessed using Student’s *t*-test, chi-square test, Mann–Whitney U or ANOVA according to the most appropriate test. Two-tailed statistical analysis was used, and *p*-values of <0.05 were considered statistically significant.

## 3. Results

### 3.1. Patient’ Characteristics

The study included 53 patients (27 males and 26 females) with a median age of 72 years. Of these, 51 received daratumumab in combination with either Len (34 patients) or Thal (15 patients), while 2 received isatuximab plus Pom. Frontline therapy was administered to 38 patients, second-line to 11, third-line to 4, and fourth-line to 1. Patient characteristics are detailed in Table 1. All patients received thromboprophylaxis per current guidelines, with 73% prescribed low dose acetylsalicylic acid (ASA).

Given the exploratory nature of this study, particular emphasis was placed on the analysis of hematological and inflammatory markers in relation to thrombotic events, which represents one of its main strengths. Given this exploratory focus, hematological and inflammatory markers potentially associated with thrombotic risk were analyzed in detail and are presented below (Section 3.4 and Section 3.5).

### 3.2. Thrombosis Risk Score Evaluation

Three conventional VTE risk scores (IMWG, IMPEDE, and SAVED) were retrospectively applied in an exploratory and descriptive manner. Due to missing cytogenetic data, the PRISM score could not be evaluated. The IMWG score classified all 49 evaluable patients as “High Risk,” mainly reflecting the intrinsic risk associated with MM diagnosis and first-line IMiD-based therapy. As shown in Figure 1, the IMPEDE score (available for 47 patients) displayed a broader distribution: 5 patients (8.9%) were categorized as High Risk, 30 (53.6%) as Intermediate Risk, and 12 (25.5%) as Low Risk. Interestingly, all thrombotic events (n = 5) occurred within the Intermediate Risk group, suggesting limited discriminatory capacity in this setting. The SAVED score classified 30 patients (56.6%) as Low Risk and 8 (15.1%) as High Risk, whereas 18 (34%) could not be scored due to incomplete data. Notably, 4 of the 5 patients who experienced thrombotic events were categorized as Low Risk, and the fifth was unclassifiable. Although purely exploratory, these descriptive observations illustrate the suboptimal performance of currently available VTE scores in real-world MM populations treated with IMiD/aCD38-based regimens, where missing data and treatment heterogeneity may affect score reliability. Overall, this analysis should be interpreted as hypothesis-generating and underscores the need for MM-specific thrombosis risk models that integrate contemporary therapeutic variables, biological risk factors, and inflammatory or immune signatures.

### 3.3. Incidence and Timing of Thrombotic Events

Thrombotic events occurred in 5 patients (9.4%), with a median time to event of 48 days from the initiation of therapy. As illustrated in Table 2, the affected patients were distributed across different demographic and treatment groups. All thrombotic events were managed effectively with LMWH, and no recurrent episodes were observed throughout the follow-up period. Remarkably, all five patients with VTE were under regular prophylaxis with ASA, according to guidelines, this raises critical concerns about ASA efficacy in preventing VTE in MM patients, particularly in those receiving immunomodulatory-based therapies.

### 3.4. Association with Hematological and Inflammatory Markers

To investigate biological markers of thrombosis, a comparative analysis of hematological and inflammatory parameters was performed between patients with (n = 5) and without (n = 48) VTE. Several biomarkers showed significant differences (Figure 2). β2M levels were markedly lower in patients who experienced TE (*p* = 0.0008), suggesting a lower tumor burden or altered clearance dynamics. Serum ferritin levels were also significantly reduced in the TE group (*p* = 0.014), indicating a diminished systemic inflammatory state. Similar trends were observed for serum intact lambda light chains (*p* = 0.017) and free lambda light chains (FLC-λ, *p* = 0.021), both of which were substantially lower in patients with TE. Furthermore, the monocyte-to-lymphocyte ratio (MLR), a known marker of systemic inflammation and immune dysregulation, was significantly reduced in TE patients (*p* = 0.017), further supporting the hypothesis that a less inflamed or immunosuppressed state may paradoxically predispose to thrombotic risk. Interestingly, only one out of five TE patients had lytic bone lesions, a common indicator of advanced disease and inflammation, suggesting that thrombotic events may arise independently of the typical high-inflammatory MM phenotype. Collectively, these data highlight a distinct biological profile associated with thrombosis in MM patients, characterized by lower inflammatory and disease activity markers.

### 3.5. Cytokine Profile and Immune Response in Thrombosis

To further characterize the immune-inflammatory milieu associated with thrombosis, cytokine profiling was conducted on plasma samples from bone marrow and peripheral blood of patients with (n = 2) and without (n = 10) TE. TE patients showed reduced expression of multiple cytokines (Figure 3). Specifically, TE patients exhibited significantly lower levels of macrophage colony-stimulating factor (M-CSF, *p* = 0.033), stem cell growth factor beta (SCGF-β), and macrophage inflammatory protein-1 alpha (MIP-1α), all of which are involved in monocyte/macrophage activation and hematopoietic support. These reductions may reflect a compromised myeloid response potentially relevant to vascular homeostasis. Conversely, granulocyte colony-stimulating factor (G-CSF) was elevated in TE patients (*p* = 0.01), indicating a potential neutrophil activation. IL-12(p40) and GM-CSF were also differentially expressed (*p* = 0.03 and *p* = 0.015, respectively), suggesting an altered immune profile.

## 4. Discussion

The findings of this study suggest the multifactorial nature of thrombotic risk in MM patients treated with IMiD/aCD38-based regimens. One of the key observations is that ASA thromboprophylaxis may not be sufficient in preventing TE in certain patients. Although based on limited numbers, the fact that all TE cases occurred despite patients were on ASA supports the need for further investigation into alternative prophylactic strategies for some individuals. The incidence of VTE in our cohort (9.4%) is consistent with previously reported rates for patients receiving IMiD-based therapy, which typically range between 5% and 12% in real-world and clinical-trial settings [10,11,12,13]. This observation indicates that the addition of anti-CD38 antibodies does not appear to increase thrombotic risk beyond that expected with IMiD-based combinations, confirming previously reported results [14].

It is important to note, however, that ASA prophylaxis was administered to all patients at treatment initiation, in accordance with IMWG and institutional guidelines, regardless of their baseline thrombotic risk profile. Therefore, the occurrence of thrombotic events in patients receiving ASA does not reflect a selection bias in prophylaxis assignment, but rather indicates that thrombotic complications may still arise despite guideline-based aspirin use. This finding should be interpreted as exploratory and hypothesis-generating, warranting further prospective investigation into the adequacy of ASA prophylaxis in the era of IMiD/aCD38-based regimens.

The identification of inflammatory and immune biomarkers associated with VTE risk may provide preliminary insights into patient stratification. Specifically, low levels of M-CSF, ferritin, and MLR were associated with VTE occurrence, suggesting that patients with a lower baseline inflammatory state might benefit more from other prophylactic measures instead of ASA. Moreover, the observed increase in G-CSF levels in VTE patients further supports the hypothesis that distinct immune alterations involving neutrophils could contribute to thrombotic risk in this patient population. These exploratory associations should be interpreted cautiously, as they might in part reflect random variation or type I statistical error.

Although our study did not directly assess underlying mechanisms, several pathways have been proposed to explain thrombotic risk in MM beyond systemic inflammation [27]. IMiDs may enhance tissue factor exposure and phosphatidylserine (PS) externalization on monocytes and endothelial cells, promoting a procoagulant state [28]. Additionally, elevated levels of coagulation factors such as FVIII and vWF, acquired resistance to activated protein C (APC), and impaired fibrinolysis have been implicated. These abnormalities can be detected through specialized assays including thrombin generation tests (TGT) [29], flow cytometry for PS+ cells, and assessment of APC sensitivity, which may offer a means for early detection of hypercoagulability in MM patients.

While prior studies have emphasized the contribution of pro-inflammatory cytokines and immune activation to thrombotic risk in MM, our exploratory data suggest that inflammation-independent mechanisms—such as endothelial or platelet activation and IMiD-related procoagulant effects—may act in parallel rather than in opposition. Therefore, inflammation and coagulation abnormalities should be considered part of an integrated, multifactorial thrombotic process in MM. These observations align with previously published data showing that high circulating levels of IL-6 and TNF-α are associated with early venous thromboembolism during IMiD-based therapy, and that both regimen composition and corticosteroid intensity significantly modulate thrombotic risk [30,31]. In this context, the lower systemic inflammation observed among patients who developed thrombosis in our cohort may reflect a distinct biological phenotype characterized by immune exhaustion or dysregulated myeloid signaling, rather than classical hyperinflammatory activation. This pattern supports the hypothesis that thrombosis in MM can arise from heterogeneous, possibly divergent, immune-inflammatory pathways.

From a translational perspective, integrating clinical and inflammatory biomarkers into risk prediction models could help refine prophylactic strategies. Multiplex cytokine profiling or simplified inflammatory indices may provide practical tools to identify patients less likely to benefit from aspirin alone and who might require early intensification with low-molecular-weight heparin or direct oral anticoagulants. Prospective multicenter validation of such biomarker-driven approaches will be crucial to enable personalized thromboprophylaxis, reduce treatment discontinuations, and ultimately improve long-term outcomes in MM.

Given these findings, there is a need to reassess thromboprophylaxis strategies in MM patients receiving IMiD/aCD38 therapy. In this context, we performed an exploratory and descriptive assessment of conventional VTE risk scores (IMWG, IMPEDE, and SAVED). While this analysis was descriptive and limited by sample size, it highlighted the suboptimal discriminatory ability of current models, with thrombotic events occurring mainly among patients categorized as low or intermediate risk. These observations further support the notion that existing scores may not fully capture MM-specific thrombotic risk under modern therapeutic regimens. However, given the small number of thrombotic events, these findings should be considered exploratory and interpreted with caution. Therefore, future prediction models should aim to incorporate inflammatory and immune biomarkers to achieve a more accurate and biologically informed risk stratification.

While these results are hypothesis-generating, they suggest the potential utility of biomarker-guided prophylactic strategies that warrant further validation. It is conceivable that anticoagulant therapies, such as direct oral anticoagulants (DOACs), could be explored for potential prophylactic benefits in high-risk patients, a hypothesis that warrants further investigation in future clinical studies

The role of inflammation in thrombosis has been increasingly recognized in the literature. Studies suggest that inflammatory cytokines such as interleukin-6 and tumor necrosis factor-alpha contribute to endothelial dysfunction and platelet activation, thereby increasing thrombotic risk [28,31]. Furthermore, alterations in the bone marrow microenvironment, including increased osteoclast activity and suppression of normal hematopoiesis, may also contribute to the prothrombotic state observed in MM patients [32]. This evidence aligns with our findings on inflammatory markers, reinforcing the biological plausibility of their association with thrombotic events.

The clinical implications of these findings could be relevant: while further validation of inflammatory biomarkers is necessary to refine thrombotic risk assessment in MM patients, large-scale prospective studies are essential to determine whether patients with specific biomarker profiles could benefit from alternative thromboprophylaxis strategies, such as DOACs. and or LMWH Additionally, a deeper understanding of the interplay between MM pathophysiology, treatment-induced immune modulation, and thrombotic risk will aid in the development of personalized management strategies for these patients.

This study has several limitations. First, it is a single-center analysis with a limited sample size and a small number of thrombotic events, which restricts the generalizability of the findings. Second, the cytokine analysis included only two patients with TE, limiting statistical power and interpretability. Consequently, the observed associations between cytokines and thrombotic risk should be regarded as preliminary and potentially influenced by sample size–related type I error. As such, these data should be viewed as exploratory and hypothesis-generating rather than conclusive. Third, the retrospective application of risk scores may have been affected by incomplete data, particularly for the PRISM models. Fourth, we acknowledge treatment heterogeneity within the cohort, as approximately one-third of patients received bortezomib in combination with thalidomide; however, since bortezomib was never administered as monotherapy but always as part of IMiD-based regimens, it is not possible to disentangle the specific contribution of each agent to thrombotic risk in this setting. The inclusion of both newly diagnosed and relapsed/refractory patients treated with heterogeneous regimens represents an additional limitation that may have influenced thrombosis risk estimation. Lastly, due to the limited availability of samples, external validation or expansion of the cohort was not feasible.

## 5. Conclusions

Despite these limitations, this study may offer important preliminary insights into the inflammatory mechanisms contributing to TE in MM patients treated with IMiD/aCD38-based regimens. However, a major challenge remains the lack of reliable predictors of thrombosis, which limits our ability to identify patients truly at risk. Equally important, the data suggest that ASA alone cannot be considered an adequate or sufficient prophylactic strategy for all patients in this setting. The observed correlations between inflammatory biomarkers and thrombotic risk underscore the urgent need for more accurate risk stratification tools and more effective, individualized thromboprophylaxis approaches. Further research is essential to validate these findings and to optimize risk assessment models, ultimately improving prevention strategies in this vulnerable patient population.

## Figures and Tables

**Figure 1 biomolecules-15-01533-f001:**
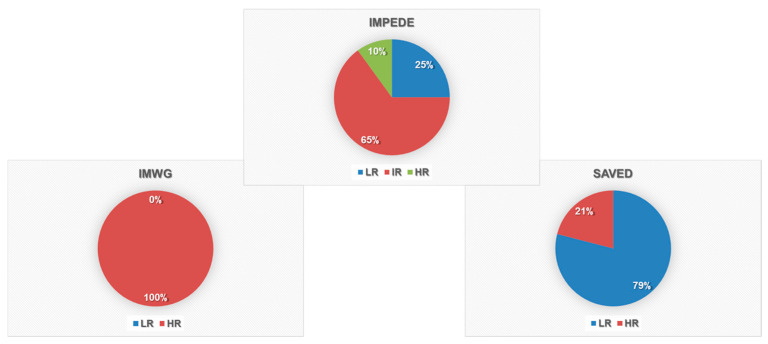
VTE risk classification according to IMWG, IMPEDE, and SAVED scores. Pie charts show the proportion of patients classified as Low Risk (LR), Intermediate Risk (IR), and High Risk (HR) using each of the three scoring systems.

**Figure 2 biomolecules-15-01533-f002:**
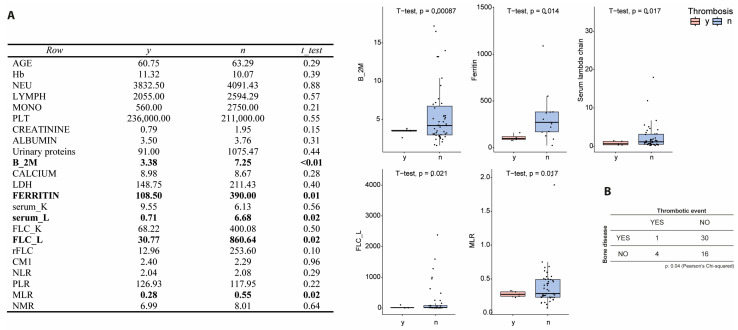
Hematological and inflammatory markers associated with thrombosis. (**A**) Boxplots comparing β2M, ferritin, serum lambda light chain, FLC lambda, and MLR between patients who developed thrombosis (y) and those who did not (n). Significant differences were observed in all five parameters. (**B**). Lytic bone lesions in patients with and without TE.

**Figure 3 biomolecules-15-01533-f003:**
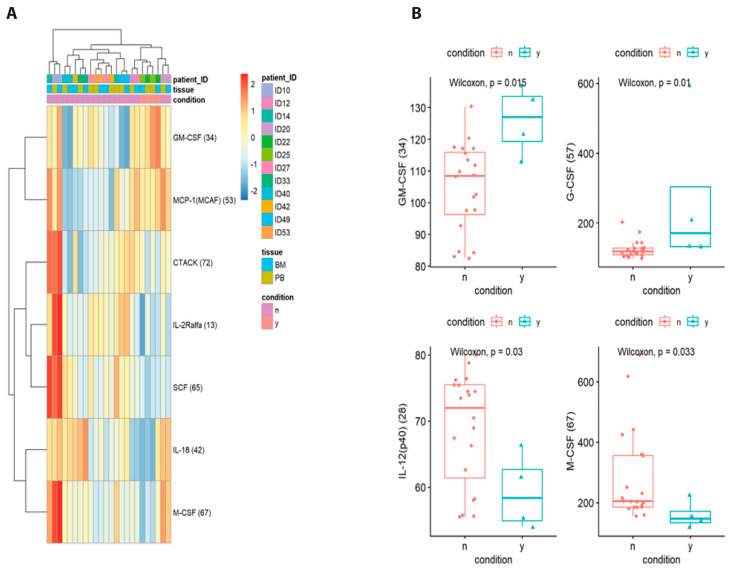
Differential cytokine expression in MM patients with and without thrombosis. (**A**) Heatmap of cytokine expression levels in peripheral blood and bone marrow samples from MM patients, stratified by thrombosis status (y/n). (**B**) Boxplots of GM−CSF, G−CSF, IL−12 (p40), and M−CSF expression levels comparing TE-positive and TE-negative patients.

**Table 1 biomolecules-15-01533-t001:** Patient characteristics.

Characteristic	Patients (n.53)
Sex (n)	M: 27 (51%) F: 26 (49%)
Age (median)	72y (47–86y)
Isotype	IgG: 37 (70%) IgA: 12 (23%) LC: 4 (7%) K: 35 (66%) L: 18 (34)
ISS (median)	ISS 2 (1–3)
Lytic lesions (n)	Y: 31 (58%) N:20 (38%)NA: 2 (4%)
Previous ASA (n)	Y: 38 (72%) N: 15 (28%)
Treatment (n)	Dara-VTD: 17 (32%) Dara-Rd: 34 (64%) Isa-Pd: 2 (4%)
Comorbidities	Cardiovascular: 15 (28%) Diabetes: 8 (15%) Chronic renal disease: 5 (9%)
ECOG	0: 19 (36%) 1: 25 (47%) 2: 6 (11%) N/A: 3 (6%)
Cytogenetic alterations	Failed or N/A: 42 (79%) SR: 5 (9%) HR: 6 (11%) (n. 3 t(4;14) and n. 3 amp 1q21)

**Table 2 biomolecules-15-01533-t002:** VTE Patient characteristics.

	VTE Pt 1	VTE Pt 2	VTE Pt 3	VTE Pt 4	VTE Pt 5
BMI > 25	Yes	Yes	Yes	No	No
Concomitant steroids *	Yes	Yes	Yes	Yes	Yes
Main comorbidities #	No	No	No	No	C/K/D
Previous TVE History	Yes	No	No	No	No
IMIDs	Len	Thal	Len	Len	Len
ASA prophylaxis	Yes	Yes	Yes	Yes	Yes
Type of TVE	SVT	DVT	SVT	DVT	SVT
Site of TVE	Saphenous collateral vein	Sural vein	Great saphenous vein	Sural vein	Saphenous collateral vein
Cycle at event onset	1	2	6	1	10
Anticoagulation Therapy	LMWH	LMWH	LMWH	LMWH	LMWH
Duration	3 m	4 m	17 m	5 m	2 m
Time to resolution (d)	35	29	63	28	29
VTE prophylaxis thereafter	Apixaban	Edoxaban	LMWH/Apixaban	Edoxaban	Apixaban

* Dexamethasone administration was 20 mg the day of Daratumumab. ^#^ C: cardiovascular; K: chronic renal disease; D: diabetes.

## Data Availability

The data supporting the conclusions of this article will be made available by the authors upon reasonable request.

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
