# Peer review of "Inflammatory Biomarkers for Thrombotic Risk Assessment in Multiple Myeloma Patients on IMiD/aCD38-Based Regimens: Insights from a Prospective Observational Study"

_biomolecules, 2025, doi:10.3390/biom15111533_

Round 1
Reviewer 1 Report
Comments and Suggestions for Authors
Botta et al. conducted a retrospective analysis using a multiplex cytokine assay and demonstrated a paradoxical relationship between thromboembolic events and inflammation in patients with multiple myeloma. The authors argue that ASA alone cannot be considered an adequate or sufficient prophylactic strategy for all patients with newly diagnosed MM receiving IMiD/aCD38-based regimens. I agree with certain aspects of the authors’ conclusions; however, several important issues remain to be addressed.
The major limitation of this study is the very small sample size and the incompleteness of the data (with missing and unclassifiable cases). Although the authors acknowledge this as a limitation, the dataset is too limited to support any definitive conclusions regarding thrombosis prevention strategies in MM. The findings can only be regarded as hypothesis-generating, and the interpretation should therefore be more cautious and toned down.
- Despite the small sample size, treatment heterogeneity is evident: one-third of patients received bortezomib in combination, yet the potential impact of bortezomib on thrombosis is not addressed.
- The cohort is too small to meaningfully evaluate the significance of conventional thrombosis risk score. Section 3.2 should be removed or moved to the Supplementary Materials, and the Introduction could instead briefly note that existing VTE risk scores may not be fully applicable in MM.
- Patients who received ASA prophylaxis were likely those whom the attending physicians considered to be at higher risk of thrombosis. The fact that all five patients who developed TE were on ASA does not provide sufficient evidence to support the claim of ASA inadequacy.
- For the prospective cohort, clarification is needed regarding which trial this refers to. Please provide the NCT number. How many patients were excluded, and for what reasons? It is also important to determine whether the reasons for exclusion were related to conditions affecting thrombosis risk. Moreover, the baseline characteristics in Table 1 should include additional data such as ECOG performance status, comorbidities, and genetic risk. Medians should be reported with ranges, and numbers should be presented as both absolute values and percentages.
In summary, given the reliance on speculation and hypothesis-driven interpretation, the discussion should be toned down. While limited by the small sample size, a notable strength of this study is the analysis of hematological and inflammatory markers in relation to thrombosis. The results should be reorganized to emphasize this aspect.
Author Response
Point by point response to reviewers’ comments
Reviewer 1
Botta et al. conducted a retrospective analysis using a multiplex cytokine assay and demonstrated a paradoxical relationship between thromboembolic events and inflammation in patients with multiple myeloma. The authors argue that ASA alone cannot be considered an adequate or sufficient prophylactic strategy for all patients with newly diagnosed MM receiving IMiD/aCD38-based regimens. I agree with certain aspects of the authors’ conclusions; however, several important issues remain to be addressed. The major limitation of this study is the very small sample size and the incompleteness of the data (with missing and unclassifiable cases). Although the authors acknowledge this as a limitation, the dataset is too limited to support any definitive conclusions regarding thrombosis prevention strategies in MM. The findings can only be regarded as hypothesis-generating, and the interpretation should therefore be more cautious and toned down.
We thank the reviewer for clearly summarizing the main points of the manuscript and we are aware of the main limitations of the study. Accordingly to suggestions, we toned down the conclusions and clearly stated the hypothesis-generating nature of the manuscript
Despite the small sample size, treatment heterogeneity is evident: one-third of patients received bortezomib in combination, yet the potential impact of bortezomib on thrombosis is not addressed.
We thank the Reviewer for this insightful comment. We have now clarified in the Discussion that bortezomib was never used as monotherapy but always in combination with thalidomide. Therefore, it is not possible to disentangle the specific contribution of bortezomib from that of the IMiD backbone regarding thrombotic risk. This limitation has been explicitly acknowledged in the revised version.
The cohort is too small to meaningfully evaluate the significance of conventional thrombosis risk score. Section 3.2 should be removed or moved to the Supplementary Materials, and the Introduction could instead briefly note that existing VTE risk scores may not be fully applicable in MM.
We thank the Reviewer for this valuable observation. We fully agree that the limited sample size precludes any statistical validation of conventional VTE risk scores. However, the intention of Section 3.2 was not to assess their predictive significance, but rather to provide a descriptive and hypothesis-generating overview of how these tools perform in a contemporary, IMiD/aCD38-treated MM population. As shown in Figure 1, all patients were classified as “high risk” by IMWG, while IMPEDE and SAVED yielded paradoxical or inconsistent risk distributions, with thrombotic events occurring mainly among patients categorized as “low” or “intermediate” risk. We believe this observation—although exploratory—illustrates the limited applicability of existing scores in real-world modern regimens, and therefore supports the rationale for developing MM-specific thrombosis risk tools. To address the Reviewer’s concern, we have now explicitly defined this analysis as exploratory in both the Results and Discussion and clarified that no inferential interpretation is intended. We would prefer to keep Figure 1 in the main text, as it visually supports this concept.
Patients who received ASA prophylaxis were likely those whom the attending physicians considered to be at higher risk of thrombosis. The fact that all five patients who developed TE were on ASA does not provide sufficient evidence to support the claim of ASA inadequacy.
We thank the Reviewer for this observation. However, in our cohort, ASA prophylaxis was uniformly prescribed to all patients initiating IMiD-based therapy, in accordance with IMWG and institutional guidelines. Therefore, the use of ASA did not reflect physician-driven risk stratification, and no selection bias based on perceived thrombotic risk was present. The fact that all five thrombotic events occurred despite ongoing ASA prophylaxis should not be interpreted as evidence of ASA inefficacy but rather as a hypothesis-generating finding suggesting that, under certain biological or inflammatory conditions, aspirin alone may be insufficient. We have clarified this point in the Discussion (page X, lines Y–Z).
For the prospective cohort, clarification is needed regarding which trial this refers to. Please provide the NCT number. How many patients were excluded, and for what reasons? It is also important to determine whether the reasons for exclusion were related to conditions affecting thrombosis risk.
We thank the Reviewer for this question and appreciate the opportunity to clarify. The present study is part of a mono-institutional observational project (both prospective and retrospective) conducted at the “Paolo Giaccone” University Hospital of Palermo. As such, it was not a clinical trial, and therefore no registration on ClinicalTrials.gov (NCT number) was required. The study was approved by the institutional ethics committee (approval numbers 05/2021 and 02/2022). Regarding inclusion and exclusion criteria, all patients with newly diagnosed multiple myeloma initiating IMiD + anti-CD38 therapy between May 2021 and December 2022 were included. Exclusion criteria were limited to (1) absence of signed informed consent and (2) presence of medical conditions that, in the investigator’s opinion, could interfere with protocol adherence or with the ability to provide informed consent. No patients were excluded for conditions related to thrombosis risk. These clarifications have been added to the Materials and Methods section.
Moreover, the baseline characteristics in Table 1 should include additional data such as ECOG performance status, comorbidities, and genetic risk. Medians should be reported with ranges, and numbers should be presented as both absolute values and percentages.
We thank the Reviewer for this valuable suggestion. Table 1 has been revised accordingly. ECOG performance status, comorbidities, and cytogenetic risk have been added to the baseline characteristics. Comorbidities are now specified as follows: cardiovascular, diabetes, and chronic renal disease. All median values are now reported together with their corresponding ranges, and all categorical variables are presented as both absolute numbers and percentages.
In summary, given the reliance on speculation and hypothesis-driven interpretation, the discussion should be toned down. While limited by the small sample size, a notable strength of this study is the analysis of hematological and inflammatory markers in relation to thrombosis. The results should be reorganized to emphasize this aspect.
We appreciate the Reviewer’s constructive feedback. The Discussion section has been revised to moderate speculative statements and to ensure that all interpretations are clearly supported by the data. We have also reorganized the Results and Discussion to emphasize the analysis of hematological and inflammatory markers in relation to thrombotic events, which now represents a key focus of the manuscript. We believe these changes improve clarity and better align the Discussion with the scope and strengths of our study.
Reviewer 2 Report
Comments and Suggestions for Authors
This is a very interesting manuscript exploring further venous thrombosis in patients with myeloma. The authors are correct that we are in need of more accurate and patient specific risk assessment tools that can capture the individual risk more effectively as thrombosis remains an important risk despite current guidelines. I do however have some important concerns regarding the content of the manuscript
- The study includes only 53 patients, with 5 thrombotic events. In addition the cytokine profiling was performed in only 2 TE cases, which is too small for meaningful conclusions. These limitations should be emphasized more strongly in the abstract and discussion as the findings (which are counterintuitive) could be attributed to the small numbers (type I error rates).
- The association betweenlower inflammatory markers and thrombosis is counterintuitive and requires a more cautious interpretation. Are the authors suggesting a distinct, non-inflammatory thrombotic pathway in MM? There are a number of reports which have explored thrombotic mechanisms in MM and report an association between proinflammatory pathways and biomarkers and thrombotic. THe authors should include these findings in the discussion in comparison/contrast to their results.
- The mechanistic explanation linking low ferritin/MLR and higher thrombosis risk is speculative. More discussion of alternative explanations (e.g., chance findings due to small sample size) is warranted
-
The emphasis on cytokine profiling is disproportionate to the very limited sample (n=2). Consider rephrasing to highlight exploratory nature
- Table 2 could include additional relevant details (e.g., concomitant corticosteroid dose, comorbidities, body mass index).
- Figures should include exact p-values rather than only stating significance
- how does the rate of VTE in the cohort compare to other cohorts in the literature that received IMiDs or IMiD/anticd38 MAbs
- To my knowledge antiCD38 antibodies have NOT been linked to increased VTE risk
- clarify that TE refers actually to venous thrombotic events and not arterial thrombosis
- Among the study limitations is the fact that the population mixed (newly diagnosed, RR and different treatment regimens)
- I am not clear which of the different scores were used to apply thromboprophylaxis
- also report whether patients received dexamethasone and what doses, dexamethasone is known to increase thombotic risk and is therefore a necessary piece of information
Author Response
Reviewer 2
This is a very interesting manuscript exploring further venous thrombosis in patients with myeloma. The authors are correct that we are in need of more accurate and patient specific risk assessment tools that can capture the individual risk more effectively as thrombosis remains an important risk despite current guidelines. I do however have some important concerns regarding the content of the manuscript.
The study includes only 53 patients, with 5 thrombotic events. In addition the cytokine profiling was performed in only 2 TE cases, which is too small for meaningful conclusions. These limitations should be emphasized more strongly in the abstract and discussion as the findings (which are counterintuitive) could be attributed to the small numbers (type I error rates).
We thank the Reviewer for this valuable observation. In response to this comment, we have further emphasized the exploratory and hypothesis-generating nature of the study. Specifically, the Abstract now explicitly states that the findings should be interpreted with caution given the small cohort and limited cytokine dataset, and the Discussion highlights that the observed associations may reflect chance findings (type I error) related to sample size. These revisions make the study’s limitations and interpretative caution more explicit, while preserving the scientific value and context of the work.
The association between lower inflammatory markers and thrombosis is counterintuitive and requires a more cautious interpretation. Are the authors suggesting a distinct, non-inflammatory thrombotic pathway in MM? There are a number of reports which have explored thrombotic mechanisms in MM and report an association between proinflammatory pathways and biomarkers and thrombotic. THe authors should include these findings in the discussion in comparison/contrast to their results.
We thank the Reviewer for this insightful comment. We agree that the association between lower inflammatory markers and thrombotic events may appear counterintuitive when compared with prior studies reporting positive correlations between inflammation and thrombosis in MM. In the revised Discussion, we have expanded the mechanistic context to explicitly acknowledge and contrast these findings. We now clarify that our results do not exclude the contribution of inflammatory pathways but rather suggest that multiple, possibly concurrent mechanisms—including inflammation, endothelial and platelet activation, and IMiD-related procoagulant effects—may underlie thrombosis in MM. The revised text also emphasizes that the observed associations should be interpreted with caution given the limited sample size and the exploratory design of the study.
The mechanistic explanation linking low ferritin/MLR and higher thrombosis risk is speculative. More discussion of alternative explanations (e.g., chance findings due to small sample size) is warranted
We thank the Reviewer for this important comment. We agree that mechanistic inferences based on the association between low ferritin/MLR and thrombosis are speculative. In the revised Discussion, we now explicitly acknowledge that these correlations may reflect random variation or chance findings due to the limited sample size and should be interpreted with caution. We have also expanded the paragraph to mention alternative explanations, including biological heterogeneity and treatment-related endothelial effects independent of inflammation.
The emphasis on cytokine profiling is disproportionate to the very limited sample (n=2). Consider rephrasing to highlight exploratory nature
We fully agree. The text referring to cytokine profiling has been rephrased throughout the Results and Discussion to clearly indicate that this analysis was exploratory and descriptive. We have also moderated the tone when discussing cytokine differences to avoid over-interpretation.
Table 2 could include additional relevant details (e.g., concomitant corticosteroid dose, comorbidities, body mass index).
We thank the Reviewer for this suggestion. Table 2 has been expanded to include data on corticosteroid exposure (dexamethasone dose at the time of the event), relevant comorbidities, and body-mass-index values for the five patients who experienced thrombotic events.
Figures should include exact p-values rather than only stating significance
We thank the Reviewer for this observation. Exact p-values are already indicated within each figure panel (e.g., “T-test, p = 0.0008”; “Wilcoxon, p = 0.015”) and described in the corresponding figure legends. We believe the apparent omission may be related to the image resolution of the PDF version reviewed. No changes were therefore required, as all exact p-values are already reported and visible in the original high-resolution figures.
how does the rate of VTE in the cohort compare to other cohorts in the literature that received IMiDs or IMiD/anticd38 Mabs
We appreciate this observation. In the revised Discussion, we now compare our VTE incidence (9.4%) with published data from real-world and clinical-trial cohorts of patients receiving IMiD-based regimens, with or without anti-CD38 monoclonal antibodies, which typically report rates between 5 % and 12 %. Our results are therefore consistent with the upper range of previously observed incidences.
To my knowledge antiCD38 antibodies have NOT been linked to increased VTE risk
We agree and have clarified this point. The revised Discussion now specify that anti-CD38 antibodies alone are not associated with increased thrombotic risk and that the observed events likely reflect the cumulative effect of IMiD-based combination therapy and patient-related risk factors.
Clarify that TE refers actually to venous thrombotic events and not arterial thrombosis
We thank the Reviewer for pointing this out. We have clarified throughout the text and in figure/table legends that TE refers specifically to venous thrombotic events (VTE).
Among the study limitations is the fact that the population mixed (newly diagnosed, RR and different treatment regimens)
We agree. This limitation has now been explicitly added to the Discussion (Limitations section), acknowledging that heterogeneity in disease stage and treatment regimens may have introduced variability in thrombotic risk assessment.
I am not clear which of the different scores were used to apply thromboprophylaxis
We appreciate this request for clarification. Thromboprophylaxis was applied according to institutional and IMWG guidelines, based on clinical risk assessment rather than formal score application. All patients received low-dose aspirin at treatment initiation, irrespective of their baseline risk profile. Patients who were already receiving anticoagulant therapy (e.g., low-molecular-weight heparin or other agents) for pre-existing indications continued their ongoing regimen without modification. This point has been clarified in the Method sections.
also report whether patients received dexamethasone and what doses, dexamethasone is known to increase thombotic risk and is therefore a necessary piece of information
We thank the Reviewer for this important comment. All patients in our cohort received dexamethasone as part of their IMiD-based combination regimen. Dexamethasone was administered weekly at a median dose of 20 mg (range 8–40 mg), adjusted according to age and comorbidities in frail patients. This information has been added to the Methods and Table 1, and the potential contribution of corticosteroid exposure to thrombotic risk is now acknowledged in the Discussion section.
Round 2
Reviewer 1 Report
Comments and Suggestions for Authors
I agree with the authors’ overall perspective, and I believe that the identified issues have been appropriately addressed, making the manuscript suitable for publication.
Reviewer 2 Report
Comments and Suggestions for Authors
I am happy with the changes and the author's responses. The authors acknowledge limitations appropriately and the manuscript is suitable for publication.